# Hierarchical Implicit Models and Likelihood-Free Variational Inference

**Dustin Tran**
Columbia University

**Rajesh Ranganath**
Princeton University

**David M. Blei**
Columbia University

## Abstract

Implicit probabilistic models are a flexible class of models defined by a simulation process for data. They form the basis for theories which encompass our understanding of the physical world. Despite this fundamental nature, the use of implicit models remains limited due to challenges in specifying complex latent structure in them, and in performing inferences in such models with large data sets. In this paper, we first introduce *hierarchical implicit model*s *(*HIMs*)*. HIMs combine the idea of implicit densities with hierarchical Bayesian modeling, thereby defining models via simulators of data with rich hidden structure. Next, we develop *likelihood-free variational inference (*LFVI*)*, a scalable variational inference algorithm for HIMs. Key to LFVI is specifying a variational family that is also implicit. This matches the model's flexibility and allows for accurate approximation of the posterior. We demonstrate diverse applications: a large-scale physical simulator for predator-prey populations in ecology; a Bayesian generative adversarial network for discrete data; and a deep implicit model for text generation.

## 1 Introduction

Consider a model of coin tosses. With probabilistic models, one typically posits a latent probability, and supposes each toss is a Bernoulli outcome given this probability [36, 15]. After observing a collection of coin tosses, Bayesian analysis lets us describe our inferences about the probability.

However, we know from the laws of physics that the outcome of a coin toss is fully determined by its initial conditions (say, the impulse and angle of flip) [25, 9]. Therefore a coin toss' randomness does not originate from a latent probability but in noisy initial parameters. This alternative model incorporates the physical system, better capturing the generative process. Furthermore the model is *implicit*, also known as a simulator: we can sample data from its generative process, but we may not have access to calculate its density [11, 20].

Coin tosses are simple, but they serve as a building block for complex implicit models. These models, which capture the laws and theories of real-world physical systems, pervade fields such as population genetics [40], statistical physics [1], and ecology [3]; they underlie structural equation models in economics and causality [39]; and they connect deeply to generative adversarial networks (GANs) [18], which use neural networks to specify a flexible implicit density [35].

Unfortunately, implicit models, including GANs, have seen limited success outside specific domains. There are two reasons. First, it is unknown how to design implicit models for more general applications, exposing rich latent structure such as priors, hierarchies, and sequences. Second, existing methods for inferring latent structure in implicit models do not sufficiently scale to high-dimensional or large data sets. In this paper, we design a new class of implicit models and we develop a new algorithm for accurate and scalable inference.

For modeling, § 2 describes *hierarchical implicit models*, a class of Bayesian hierarchical models which only assume a process that generates samples. This class encompasses both simulators in the

classical literature and those employed in GANs. For example, we specify a Bayesian GAN, where we place a prior on its parameters. The Bayesian perspective allows GANs to quantify uncertainty and improve data efficiency. We can also apply them to discrete data; this setting is not possible with traditional estimation algorithms for GANs [27].

For inference, § 3 develops *likelihood-free variational inference (*LFVI*)*, which combines variational inference with density ratio estimation [49, 35]. Variational inference posits a family of distributions over latent variables and then optimizes to find the member closest to the posterior [23]. Traditional approaches require a likelihood-based model and use crude approximations, employing a simple approximating family for fast computation. LFVI expands variational inference to implicit models and enables accurate variational approximations with implicit variational families: LFVI does not require the variational density to be tractable. Further, unlike previous Bayesian methods for implicit models, LFVI scales to millions of data points with stochastic optimization.

This work has diverse applications. First, we analyze a classical problem from the approximate Bayesian computation (ABC) literature, where the model simulates an ecological system [3]. We analyze 100,000 time series which is not possible with traditional methods. Second, we analyze a Bayesian GAN, which is a GAN with a prior over its weights. Bayesian GANs outperform corresponding Bayesian neural networks with known likelihoods on several classification tasks. Third, we show how injecting noise into hidden units of recurrent neural networks corresponds to a deep implicit model for flexible sequence generation.

**Related Work.**   This paper connects closely to three lines of work. The first is Bayesian inference for implicit models, known in the statistics literature as approximate Bayesian computation (ABC) [3, 33]. ABC steps around the intractable likelihood by applying summary statistics to measure the closeness of simulated samples to real observations. While successful in many domains, ABC has shortcomings. First, the results generated by ABC depend heavily on the chosen summary statistics and the closeness measure. Second, as the dimensionality grows, closeness becomes harder to achieve. This is the classic curse of dimensionality.

The second is GANs [18]. GANs have seen much interest since their conception, providing an efficient method for estimation in neural network-based simulators. Larsen et al. [28] propose a hybrid of variational methods and GANs for improved reconstruction. Chen et al. [7] apply information penalties to disentangle factors of variation. Donahue et al. [12], Dumoulin et al. [13] propose to match on an augmented space, simultaneously training the model and an inverse mapping from data to noise. Unlike any of the above, we develop models with explicit priors on latent variables, hierarchies, and sequences, and we generalize GANs to perform Bayesian inference.

The final thread is variational inference with expressive approximations [45, 48, 52]. The idea of casting the design of variational families as a modeling problem was proposed in Ranganath et al. [44]. Further advances have analyzed variational programs [42]—a family of approximations which only requires a process returning samples—and which has seen further interest [30]. Implicit-like variational approximations have also appeared in auto-encoder frameworks [32, 34] and message passing [24]. We build on variational programs for inferring implicit models.

## 2   Hierarchical Implicit Models

Hierarchical models play an important role in sharing statistical strength across examples [16]. For a broad class of hierarchical Bayesian models, the joint distribution of the hidden and observed variables is

$$p(\mathbf{x}, \mathbf{z}, \boldsymbol{\beta}) = p(\boldsymbol{\beta}) \prod_{n=1}^{N} p(\mathbf{x}_n \,|\, \mathbf{z}_n, \boldsymbol{\beta}) p(\mathbf{z}_n \,|\, \boldsymbol{\beta}), \tag{1}$$

where $\mathbf{x}_n$ is an observation, $\mathbf{z}_n$ are latent variables associated to that observation (local variables), and $\boldsymbol{\beta}$ are latent variables shared across observations (global variables). See Fig. 1 (left).

With hierarchical models, local variables can be used for clustering in mixture models, mixed memberships in topic models [4], and factors in probabilistic matrix factorization [47]. Global variables can be used to pool information across data points for hierarchical regression [16], topic models [4], and Bayesian nonparametrics [50].

Hierarchical models typically use a tractable likelihood $p(\mathbf{x}_n \,|\, \mathbf{z}_n, \boldsymbol{\beta})$.  But many likelihoods of interest, such as simulator-based models [20] and generative adversarial networks [18], admit high

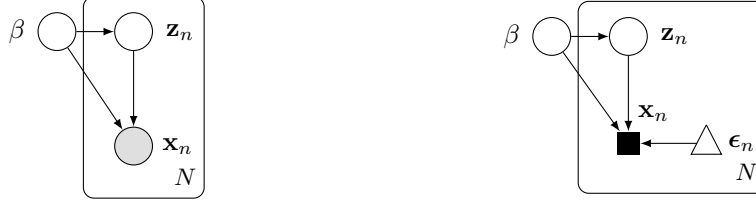

**Figure 1:** (**left**) Hierarchical model, with local variables $\mathbf{z}$ and global variables $\beta$. (**right**) **Hierarchical implicit model**. It is a hierarchical model where $\mathbf{x}$ is a deterministic function (denoted with a square) of noise $\boldsymbol{\epsilon}$ (denoted with a triangle).

fidelity to the true data generating process and do not admit a tractable likelihood. To overcome this limitation, we develop *hierarchical implicit models* (HIMs).

Hierarchical implicit models have the same joint factorization as Eq.1 but only assume that one can sample from the likelihood. Rather than define $p(\mathbf{x}_n \mid \mathbf{z}_n, \boldsymbol{\beta})$ explicitly, HIMs define a function $g$ that takes in random noise $\boldsymbol{\epsilon}_n \sim s(\cdot)$ and outputs $\mathbf{x}_n$ given $\mathbf{z}_n$ and $\boldsymbol{\beta}$,

$$\mathbf{x}_n = g(\boldsymbol{\epsilon}_n \mid \mathbf{z}_n, \boldsymbol{\beta}), \quad \boldsymbol{\epsilon}_n \sim s(\cdot).$$

The induced, implicit likelihood of $\mathbf{x}_n \in A$ given $\mathbf{z}_n$ and $\boldsymbol{\beta}$ is

$$\mathcal{P}(\mathbf{x}_n \in A \mid \mathbf{z}_n, \boldsymbol{\beta}) = \int_{\{g(\boldsymbol{\epsilon}_n \mid \mathbf{z}_n, \boldsymbol{\beta}) = \mathbf{x}_n \in A\}} s(\boldsymbol{\epsilon}_n) \, \mathrm{d}\boldsymbol{\epsilon}_n.$$

This integral is typically intractable. It is difficult to find the set to integrate over, and the integration itself may be expensive for arbitrary noise distributions $s(\cdot)$ and functions $g$.

Fig. 1 (right) displays the graphical model for HIMs. Noise ($\boldsymbol{\epsilon}_n$) are denoted by triangles; deterministic computation ($\mathbf{x}_n$) are denoted by squares. We illustrate two examples.

**Example: Physical Simulators.** Given initial conditions, simulators describe a stochastic process that generates data. For example, in population ecology, the Lotka-Volterra model simulates predator-prey populations over time via a stochastic differential equation [55]. For prey and predator populations $x_1, x_2 \in \mathbb{R}^+$ respectively, one process is

$$\frac{\mathrm{d}x_1}{\mathrm{d}t} = \beta_1 x_1 - \beta_2 x_1 x_2 + \epsilon_1, \qquad \epsilon_1 \sim \mathrm{Normal}(0, 10),$$

$$\frac{\mathrm{d}x_2}{\mathrm{d}t} = -\beta_2 x_2 + \beta_3 x_1 x_2 + \epsilon_2, \quad \epsilon_2 \sim \mathrm{Normal}(0, 10),$$

where Gaussian noises $\epsilon_1, \epsilon_2$ are added at each full time step. The simulator runs for $T$ time steps given initial population sizes for $x_1, x_2$. Lognormal priors are placed over $\beta$. The Lotka-Volterra model is grounded by theory but features an intractable likelihood. We study it in § 4.

**Example: Bayesian Generative Adversarial Network.** Generative adversarial networks (GANs) define an implicit model and a method for parameter estimation [18]. They are known to perform well on image generation [41]. Formally, the implicit model for a GAN is

$$\mathbf{x}_n = g(\boldsymbol{\epsilon}_n; \boldsymbol{\theta}), \quad \boldsymbol{\epsilon}_n \sim s(\cdot), \tag{2}$$

where $g$ is a neural network with parameters $\boldsymbol{\theta}$, and $s$ is a standard normal or uniform. The neural network $g$ is typically not invertible; this makes the likelihood intractable.

The parameters $\boldsymbol{\theta}$ in GANs are estimated by divergence minimization between the generated and real data. We make GANs amenable to Bayesian analysis by placing a prior on the parameters $\boldsymbol{\theta}$. We call this a Bayesian GAN. Bayesian GANs enable modeling of parameter uncertainty and are inspired by Bayesian neural networks, which have been shown to improve the uncertainty and data efficiency of standard neural networks [31, 37]. We study Bayesian GANs in § 4; Appendix B provides example implementations in the Edward probabilistic programming language [53].

## 3 Likelihood-Free Variational Inference

We described hierarchical implicit models, a rich class of latent variable models with local and global structure alongside an implicit density. Given data, we aim to calculate the model's posterior $p(\mathbf{z}, \boldsymbol{\beta} \mid \mathbf{x}) = p(\mathbf{x}, \mathbf{z}, \boldsymbol{\beta})/p(\mathbf{x})$. This is difficult as the normalizing constant $p(\mathbf{x})$ is typically

intractable. With implicit models, the lack of a likelihood function introduces an additional source of intractability.

We use variational inference [23]. It posits an approximating family $q \in \mathcal{Q}$ and optimizes to find the member closest to $p(\mathbf{z}, \boldsymbol{\beta} \,|\, \mathbf{x})$. There are many choices of variational objectives that measure closeness [42, 29, 10]. To choose an objective, we lay out desiderata for a variational inference algorithm for implicit models:

1. *Scalability*. Machine learning hinges on stochastic optimization to scale to massive data [6]. The variational objective should admit unbiased subsampling with the standard technique,

$$ \sum_{n=1}^{N} f(\mathbf{x}_n) \approx \frac{N}{M} \sum_{m=1}^{M} f(\mathbf{x}_m), $$

where some computation $f(\cdot)$ over the full data is approximated with a mini-batch of data $\{\mathbf{x}_m\}$.

2. *Implicit Local Approximations*. Implicit models specify flexible densities; this induces very complex posterior distributions. Thus we would like a rich approximating family for the per-data point approximations $q(\mathbf{z}_n \,|\, \mathbf{x}_n, \boldsymbol{\beta})$. This means the variational objective should only require that one can sample $\mathbf{z}_n \sim q(\mathbf{z}_n \,|\, \mathbf{x}_n, \boldsymbol{\beta})$ and not evaluate its density.

One variational objective meeting our desiderata is based on the classical minimization of the Kullback-Leibler (KL) divergence. (Surprisingly, Appendix C details how the KL is the *only* possible objective among a broad class.)

## 3.1 KL Variational Objective

Classical variational inference minimizes the KL divergence from the variational approximation $q$ to the posterior. This is equivalent to maximizing the *evidence lower bound* (ELBO),

$$ \mathcal{L} = \mathbb{E}_{q(\boldsymbol{\beta}, \mathbf{z} \,|\, \mathbf{x})}[\log p(\mathbf{x}, \mathbf{z}, \boldsymbol{\beta}) - \log q(\boldsymbol{\beta}, \mathbf{z} \,|\, \mathbf{x})]. \tag{3} $$

Let $q$ factorize in the same way as the posterior,

$$ q(\boldsymbol{\beta}, \mathbf{z} \,|\, \mathbf{x}) = q(\boldsymbol{\beta}) \prod_{n=1}^{N} q(\mathbf{z}_n \,|\, \mathbf{x}_n, \boldsymbol{\beta}), $$

where $q(\mathbf{z}_n \,|\, \mathbf{x}_n, \boldsymbol{\beta})$ is an intractable density and since the data $\mathbf{x}$ is constant during inference, we drop conditioning for the global $q(\boldsymbol{\beta})$. Substituting $p$ and $q$'s factorization yields

$$ \mathcal{L} = \mathbb{E}_{q(\boldsymbol{\beta})}[\log p(\boldsymbol{\beta}) - \log q(\boldsymbol{\beta})] + \sum_{n=1}^{N} \mathbb{E}_{q(\boldsymbol{\beta})q(\mathbf{z}_n \,|\, \mathbf{x}_n, \boldsymbol{\beta})}[\log p(\mathbf{x}_n, \mathbf{z}_n \,|\, \boldsymbol{\beta}) - \log q(\mathbf{z}_n \,|\, \mathbf{x}_n, \boldsymbol{\beta})]. $$

This objective presents difficulties: the local densities $p(\mathbf{x}_n, \mathbf{z}_n \,|\, \boldsymbol{\beta})$ and $q(\mathbf{z}_n \,|\, \mathbf{x}_n, \boldsymbol{\beta})$ are both intractable. To solve this, we consider ratio estimation.

## 3.2 Ratio Estimation for the KL Objective

Let $q(\mathbf{x}_n)$ be the empirical distribution on the observations $\mathbf{x}$ and consider using it in a "variational joint" $q(\mathbf{x}_n, \mathbf{z}_n \,|\, \boldsymbol{\beta}) = q(\mathbf{x}_n)q(\mathbf{z}_n \,|\, \mathbf{x}_n, \boldsymbol{\beta})$. Now subtract the log empirical $\log q(\mathbf{x}_n)$ from the ELBO above. The ELBO reduces to

$$ \mathcal{L} \propto \mathbb{E}_{q(\boldsymbol{\beta})}[\log p(\boldsymbol{\beta}) - \log q(\boldsymbol{\beta})] + \sum_{n=1}^{N} \mathbb{E}_{q(\boldsymbol{\beta})q(\mathbf{z}_n \,|\, \mathbf{x}_n, \boldsymbol{\beta})} \left[ \log \frac{p(\mathbf{x}_n, \mathbf{z}_n \,|\, \boldsymbol{\beta})}{q(\mathbf{x}_n, \mathbf{z}_n \,|\, \boldsymbol{\beta})} \right]. \tag{4} $$

(Here the proportionality symbol means equality up to additive constants.) Thus the ELBO is a function of the ratio of two intractable densities. If we can form an estimator of this ratio, we can proceed with optimizing the ELBO.

We apply techniques for ratio estimation [49]. It is a key idea in GANs [35, 54], and similar ideas have rearisen in statistics and physics [19, 8]. In particular, we use class probability estimation: given a sample from $p(\cdot)$ or $q(\cdot)$ we aim to estimate the probability that it belongs to $p(\cdot)$. We model

this using $\sigma(r(\cdot; \boldsymbol{\theta}))$, where $r$ is a parameterized function (e.g., neural network) taking sample inputs and outputting a real value; $\sigma$ is the logistic function outputting the probability.

We train $r(\cdot; \boldsymbol{\theta})$ by minimizing a loss function known as a proper scoring rule [17]. For example, in experiments we use the log loss,

$$\mathcal{D}_{\log} = \mathbb{E}_{p(\mathbf{x}_n, \mathbf{z}_n \mid \boldsymbol{\beta})}[-\log \sigma(r(\mathbf{x}_n, \mathbf{z}_n, \boldsymbol{\beta}; \boldsymbol{\theta}))] + \mathbb{E}_{q(\mathbf{x}_n, \mathbf{z}_n \mid \boldsymbol{\beta})}[-\log(1 - \sigma(r(\mathbf{x}_n, \mathbf{z}_n, \boldsymbol{\beta}; \boldsymbol{\theta})))]. \quad (5)$$

The loss is zero if $\sigma(r(\cdot; \boldsymbol{\theta}))$ returns 1 when a sample is from $p(\cdot)$ and 0 when a sample is from $q(\cdot)$. (We also experiment with the hinge loss; see § 4.) If $r(\cdot; \boldsymbol{\theta})$ is sufficiently expressive, minimizing the loss returns the optimal function [35],

$$r^*(\mathbf{x}_n, \mathbf{z}_n, \boldsymbol{\beta}) = \log p(\mathbf{x}_n, \mathbf{z}_n \mid \boldsymbol{\beta}) - \log q(\mathbf{x}_n, \mathbf{z}_n \mid \boldsymbol{\beta}).$$

As we minimize Eq.5, we use $r(\cdot; \boldsymbol{\theta})$ as a proxy to the log ratio in Eq.4. Note $r$ estimates the log ratio; it's of direct interest and more numerically stable than the ratio.

The gradient of $\mathcal{D}_{\log}$ with respect to $\boldsymbol{\theta}$ is

$$\mathbb{E}_{p(\mathbf{x}_n, \mathbf{z}_n \mid \boldsymbol{\beta})}[\nabla_{\boldsymbol{\theta}} \log \sigma(r(\mathbf{x}_n, \mathbf{z}_n, \boldsymbol{\beta}; \boldsymbol{\theta}))] + \mathbb{E}_{q(\mathbf{x}_n, \mathbf{z}_n \mid \boldsymbol{\beta})}[\nabla_{\boldsymbol{\theta}} \log(1 - \sigma(r(\mathbf{x}_n, \mathbf{z}_n, \boldsymbol{\beta}; \boldsymbol{\theta})))]. \quad (6)$$

We compute unbiased gradients with Monte Carlo.

## 3.3 Stochastic Gradients of the KL Objective

To optimize the ELBO, we use the ratio estimator,

$$\mathcal{L} = \mathbb{E}_{q(\boldsymbol{\beta} \mid \mathbf{x})}[\log p(\boldsymbol{\beta}) - \log q(\boldsymbol{\beta})] + \sum_{n=1}^{N} \mathbb{E}_{q(\boldsymbol{\beta} \mid \mathbf{x}) q(\mathbf{z}_n \mid \mathbf{x}_n, \boldsymbol{\beta})}[r(\mathbf{x}_n, \mathbf{z}_n, \boldsymbol{\beta})]. \quad (7)$$

All terms are now tractable. We can calculate gradients to optimize the variational family $q$. Below we assume the priors $p(\boldsymbol{\beta}), p(\mathbf{z}_n \mid \boldsymbol{\beta})$ are differentiable. (We discuss methods to handle discrete global variables in the next section.)

We focus on reparameterizable variational approximations [26, 46]. They enable sampling via a differentiable transformation $T$ of random noise, $\delta \sim s(\cdot)$. Due to Eq.7, we require the global approximation $q(\boldsymbol{\beta}; \boldsymbol{\lambda})$ to admit a tractable density. With reparameterization, its sample is

$$\boldsymbol{\beta} = T_{\text{global}}(\boldsymbol{\delta}_{\text{global}}; \boldsymbol{\lambda}), \quad \boldsymbol{\delta}_{\text{global}} \sim s(\cdot),$$

for a choice of transformation $T_{\text{global}}(\cdot; \boldsymbol{\lambda})$ and noise $s(\cdot)$. For example, setting $s(\cdot) = \mathcal{N}(0, 1)$ and $T_{\text{global}}(\boldsymbol{\delta}_{\text{global}}) = \mu + \sigma \boldsymbol{\delta}_{\text{global}}$ induces a normal distribution $\mathcal{N}(\mu, \sigma^2)$.

Similarly for the local variables $\mathbf{z}_n$, we specify

$$\mathbf{z}_n = T_{\text{local}}(\boldsymbol{\delta}_n, \mathbf{x}_n, \boldsymbol{\beta}; \boldsymbol{\phi}), \quad \boldsymbol{\delta}_n \sim s(\cdot).$$

Unlike the global approximation, the local variational density $q(\mathbf{z}_n \mid \mathbf{x}_n; \boldsymbol{\phi})$ need not be tractable: the ratio estimator relaxes this requirement. It lets us leverage implicit models not only for data but also for approximate posteriors. In practice, we also amortize computation with inference networks, sharing parameters $\boldsymbol{\phi}$ across the per-data point approximate posteriors.

The gradient with respect to global parameters $\boldsymbol{\lambda}$ under this approximating family is

$$\nabla_{\boldsymbol{\lambda}} \mathcal{L} = \mathbb{E}_{s(\boldsymbol{\delta}_{\text{global}})}[\nabla_{\boldsymbol{\lambda}}(\log p(\boldsymbol{\beta}) - \log q(\boldsymbol{\beta}))]] + \sum_{n=1}^{N} \mathbb{E}_{s(\boldsymbol{\delta}_{\text{global}}) s_n(\boldsymbol{\delta}_n)}[\nabla_{\boldsymbol{\lambda}} r(\mathbf{x}_n, \mathbf{z}_n, \boldsymbol{\beta})]. \quad (8)$$

The gradient backpropagates through the local sampling $\mathbf{z}_n = T_{\text{local}}(\boldsymbol{\delta}_n, \mathbf{x}_n, \boldsymbol{\beta}; \boldsymbol{\phi})$ and the global reparameterization $\boldsymbol{\beta} = T_{\text{global}}(\boldsymbol{\delta}_{\text{global}}; \boldsymbol{\lambda})$. We compute unbiased gradients with Monte Carlo. The gradient with respect to local parameters $\boldsymbol{\phi}$ is

$$\nabla_{\boldsymbol{\phi}} \mathcal{L} = \sum_{n=1}^{N} \mathbb{E}_{q(\boldsymbol{\beta}) s(\boldsymbol{\delta}_n)}[\nabla_{\boldsymbol{\phi}} r(\mathbf{x}_n, \mathbf{z}_n, \boldsymbol{\beta})]. \quad (9)$$

where the gradient backpropagates through $T_{\text{local}}$.[1]

**Algorithm 1:** Likelihood-free variational inference (LFVI)

---

**Input** : Model $\mathbf{x}_n, \mathbf{z}_n \sim p(\cdot \,|\, \boldsymbol{\beta}), p(\boldsymbol{\beta})$
         Variational approximation $\mathbf{z}_n \sim q(\cdot \,|\, \mathbf{x}_n, \boldsymbol{\beta}; \boldsymbol{\phi}), q(\boldsymbol{\beta} \,|\, \mathbf{x}; \boldsymbol{\lambda})$,
         Ratio estimator $r(\cdot; \boldsymbol{\theta})$
**Output:** Variational parameters $\boldsymbol{\lambda}, \boldsymbol{\phi}$
Initialize $\boldsymbol{\theta}, \boldsymbol{\lambda}, \boldsymbol{\phi}$ randomly.
**while** *not converged* **do**
    Compute unbiased estimate of $\nabla_{\boldsymbol{\theta}} \mathcal{D}$ (Eq.6), $\nabla_{\boldsymbol{\lambda}} \mathcal{L}$ (Eq.8), $\nabla_{\boldsymbol{\phi}} \mathcal{L}$ (Eq.9).
    Update $\boldsymbol{\theta}, \boldsymbol{\lambda}, \boldsymbol{\phi}$ using stochastic gradient descent.
**end**

---

### 3.4 Algorithm

Algorithm 1 outlines the procedure. We call it *likelihood-free variational inference (*LFVI*)*. LFVI is black box: it applies to models in which one can simulate data and local variables, and calculate densities for the global variables. LFVI first updates $\boldsymbol{\theta}$ to improve the ratio estimator $r$. Then it uses $r$ to update parameters $\{\boldsymbol{\lambda}, \boldsymbol{\phi}\}$ of the variational approximation $q$. We optimize $r$ and $q$ simultaneously. The algorithm is available in Edward [53].

LFVI is scalable: we can unbiasedly estimate the gradient over the full data set with mini-batches [22]. The algorithm can also handle models of either continuous or discrete data. The requirement for differentiable global variables and reparameterizable global approximations can be relaxed using score function gradients [43].

Point estimates of the global parameters $\boldsymbol{\beta}$ suffice for many applications [18, 46]. Algorithm 1 can find point estimates: place a point mass approximation $q$ on the parameters $\boldsymbol{\beta}$. This simplifies gradients and corresponds to variational EM.

## 4 Experiments

We developed new models and inference. For experiments, we study three applications: a large-scale physical simulator for predator-prey populations in ecology; a Bayesian GAN for supervised classification; and a deep implicit model for symbol generation. In addition, Appendix F, provides practical advice on how to address the stability of the ratio estimator by analyzing a toy experiment. We initialize parameters from a standard normal and apply gradient descent with ADAM.

**Lotka-Volterra Predator-Prey Simulator.** We analyze the Lotka-Volterra simulator of § 2 and follow the same setup and hyperparameters of Papamakarios and Murray [38]. Its global variables $\boldsymbol{\beta}$ govern rates of change in a simulation of predator-prey populations. To infer them, we posit a mean-field normal approximation (reparameterized to be on the same support) and run Algorithm 1 with both a log loss and hinge loss for the ratio estimation problem; Appendix D details the hinge loss. We compare to rejection ABC, MCMC-ABC, and SMC-ABC [33]. MCMC-ABC uses a spherical Gaussian proposal; SMC-ABC is manually tuned with a decaying epsilon schedule; all ABC methods are tuned to use the best performing hyperparameters such as the tolerance error.

Fig. 2 displays results on two data sets. In the top figures and bottom left, we analyze data consisting of a simulation for $T = 30$ time steps, with recorded values of the populations every $0.2$ time units. The bottom left figure calculates the negative log probability of the true parameters over the tolerance error for ABC methods; smaller tolerances result in more accuracy but slower runtime. The top figures compare the marginal posteriors for two parameters using the smallest tolerance for the ABC methods. Rejection ABC, MCMC-ABC, and SMC-ABC all contain the true parameters in their 95% credible interval but are less confident than our methods. Further, they required $100,000$ simulations from the model, with an acceptance rate of $0.004\%$ and $2.990\%$ for rejection ABC and MCMC-ABC respectively.

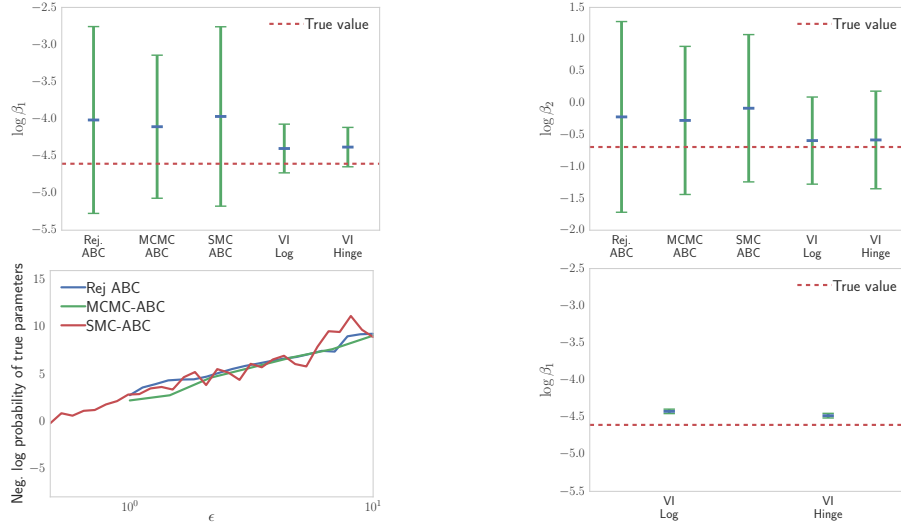

**Figure 2: (top)** Marginal posterior for first two parameters. **(bot. left)** ABC methods over tolerance error. **(bot. right)** Marginal posterior for first parameter on a large-scale data set. Our inference achieves more accurate results and scales to massive data.

|                       |       | Test Set Error |           |        |
| Model + Inference     | Crabs | Pima           | Covertype | MNIST  |
| --------------------- | ----- | -------------- | --------- | ------ |
| Bayesian GAN + VI     | 0.03  | **0.232**      | **0.154** | **0.0136** |
| Bayesian GAN + MAP    | 0.12  | 0.240          | 0.185     | 0.0283 |
| Bayesian NN + VI      | **0.02** | 0.242       | 0.164     | 0.0311 |
| Bayesian NN + MAP     | 0.05  | 0.320          | 0.188     | 0.0623 |

**Table 1:** Classification accuracy of Bayesian GAN and Bayesian neural networks across small to medium-size data sets. Bayesian GANs achieve comparable or better performance to their Bayesian neural net counterpart.

The bottom right figure analyzes data consisting of $100,000$ time series, each of the same size as the single time series analyzed in the previous figures. This size is not possible with traditional methods. Further, we see that with our methods, the posterior concentrates near the truth. We also experienced little difference in accuracy between using the log loss or the hinge loss for ratio estimation.

**Bayesian Generative Adversarial Networks.** We analyze Bayesian GANs, described in § 2. Mimicking a use case of Bayesian neural networks [5, 21], we apply Bayesian GANs for classification on small to medium-size data. The GAN defines a conditional $p(y_n \mid \mathbf{x}_n)$, taking a feature $\mathbf{x}_n \in \mathbb{R}^D$ as input and generating a label $y_n \in \{1, \dots, K\}$, via the process

$$y_n = g(\mathbf{x}_n, \boldsymbol{\epsilon}_n \mid \boldsymbol{\theta}), \qquad \boldsymbol{\epsilon}_n \sim \mathcal{N}(0, 1), \tag{10}$$

where $g(\cdot \mid \boldsymbol{\theta})$ is a 2-layer multilayer perception with ReLU activations, batch normalization, and is parameterized by weights and biases $\boldsymbol{\theta}$. We place normal priors, $\boldsymbol{\theta} \sim \mathcal{N}(0, 1)$.

We analyze two choices of the variational model: one with a mean-field normal approximation for $q(\boldsymbol{\theta} \mid \mathbf{x})$, and another with a point mass approximation (equivalent to maximum a posteriori). We compare to a Bayesian neural network, which uses the same generative process as Eq.10 but draws from a Categorical distribution rather than feeding noise into the neural net. We fit it separately using a mean-field normal approximation and maximum a posteriori. Table 1 shows that Bayesian GANs generally outperform their Bayesian neural net counterpart.

Note that Bayesian GANs can analyze discrete data such as in generating a classification label. Traditional GANs for discrete data is an open challenge [27]. In Appendix E, we compare Bayesian GANs with point estimation to typical GANs. Bayesian GANs are also able to leverage parameter uncertainty for analyzing these small to medium-size data sets.

One problem with Bayesian GANs is that they cannot work with very large neural networks: the ratio estimator is a function of global parameters, and thus the input size grows with the size of the

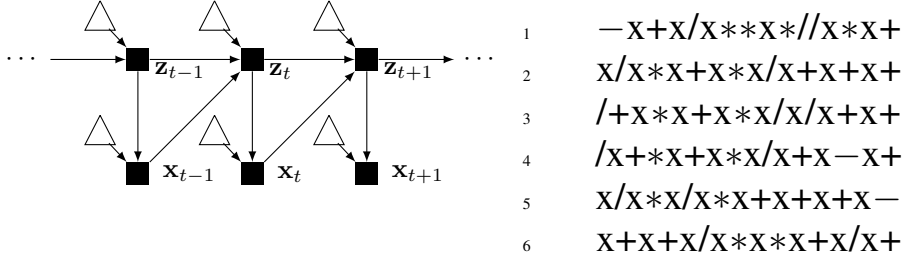

<div style="display:flex">
<div>

**(a)** A deep implicit model for sequences. It is a recurrent neural network (RNN) with noise injected into each hidden state. The hidden state is now an implicit latent variable. The same occurs for generating outputs.

</div>
<div>

**(b)** Generated symbols from the implicit model. Good samples place arithmetic operators between the variable $x$. The implicit model learned to follow rules from the context free grammar up to some multiple operator repeats.

</div>
</div>

neural network. One approach is to make the ratio estimator not a function of the global parameters. Instead of optimizing model parameters via variational EM, we can train the model parameters by backpropagating through the ratio objective instead of the variational objective. An alternative is to use the hidden units as input which is much lower dimensional [51, Appendix C].

**Injecting Noise into Hidden Units.** In this section, we show how to build a hierarchical implicit model by simply injecting randomness into hidden units. We model sequences $\mathbf{x} = (\mathbf{x}_1, \ldots, \mathbf{x}_T)$ with a recurrent neural network. For $t = 1, \ldots, T$,

$$\mathbf{z}_t = g_z(\mathbf{x}_{t-1}, \mathbf{z}_{t-1}, \boldsymbol{\epsilon}_{t,z}), \quad \boldsymbol{\epsilon}_{t,z} \sim \mathcal{N}(0,1),$$
$$\mathbf{x}_t = g_x(\mathbf{z}_t, \boldsymbol{\epsilon}_{t,x}), \quad \boldsymbol{\epsilon}_{t,x} \sim \mathcal{N}(0,1),$$

where $g_z$ and $g_x$ are both 1-layer multilayer perceptions with ReLU activation and layer normalization. We place standard normal priors over all weights and biases. See Fig. 3a.

If the injected noise $\boldsymbol{\epsilon}_{t,z}$ combines linearly with the output of $g_z$, the induced distribution $p(\mathbf{z}_t \mid \mathbf{x}_{t-1}, \mathbf{z}_{t-1})$ is Gaussian parameterized by that output. This defines a stochastic RNN [2, 14], which generalizes its deterministic connection. With nonlinear combinations, the implicit density is more flexible (and intractable), making previous methods for inference not applicable. In our method, we perform variational inference and specify $q$ to be implicit; we use the same architecture as the probability model's implicit priors.

We follow the same setup and hyperparameters as Kusner and Hernández-Lobato [27] and generate simple one-variable arithmetic sequences following a context free grammar,

$$S \to x \| S + S \| S - S \| S * S \| S / S,$$

where $\|$ divides possible productions of the grammar. We concatenate the inputs and point estimate the global variables (model parameters) using variational EM. Fig. 3b displays samples from the inferred model, training on sequences with a maximum of 15 symbols. It achieves sequences which roughly follow the context free grammar.

## 5   Discussion

We developed a class of hierarchical implicit models and likelihood-free variational inference, merging the idea of implicit densities with hierarchical Bayesian modeling and approximate posterior inference. This expands Bayesian analysis with the ability to apply neural samplers, physical simulators, and their combination with rich, interpretable latent structure.

More stable inference with ratio estimation is an open challenge. This is especially important when we analyze large-scale real world applications of implicit models. Recent work for genomics offers a promising solution [51].

**Acknowledgements.**   We thank Balaji Lakshminarayanan for discussions which helped motivate this work. We also thank Christian Naesseth, Jaan Altosaar, and Adji Dieng for their feedback and comments.  DT is supported by a Google Ph.D. Fellowship in Machine Learning and an Adobe Research Fellowship. This work is also supported by NSF IIS-0745520, IIS-1247664, IIS-1009542, ONR N00014-11-1-0651, DARPA FA8750-14-2-0009, N66001-15-C-4032, Facebook, Adobe, Amazon, and the John Templeton Foundation.

## Footnotes

[1]The ratio $r$ indirectly depends on $\boldsymbol{\phi}$ but its gradient w.r.t. $\boldsymbol{\phi}$ disappears. This is derived via the score function identity and the product rule (see, e.g., Ranganath et al. [43, Appendix]).

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
