[Supplementary Material]

# Supplement: Hierarchical Implicit Models and Likelihood-Free Variational Inference

**Dustin Tran**
Columbia University

**Rajesh Ranganath**
Princeton University

**David M. Blei**
Columbia University

## A    Noise versus Latent Variables

Hierarchical implicit models (HIMs) have two sources of randomness for each data point: the latent variable $\mathbf{z}_n$ and the noise $\boldsymbol{\epsilon}_n$; these sources of randomness get transformed to produce $\mathbf{x}_n$. Bayesian analysis infers posteriors on latent variables. A natural question is whether one should also infer the posterior of the noise.

The posterior's shape—and ultimately if it is meaningful—is determined by the dimensionality of noise and the transformation. For example, consider the generative adversarial network (GAN) model, which has no local latent variable, $\mathbf{x}_n = g(\boldsymbol{\epsilon}_n; \boldsymbol{\theta})$. The conditional $p(\mathbf{x}_n \,|\, \boldsymbol{\epsilon}_n)$ is a point mass, fully determined by $\boldsymbol{\epsilon}_n$. When $g(\cdot; \boldsymbol{\theta})$ is injective, the posterior $p(\boldsymbol{\epsilon}_n \,|\, \mathbf{x}_n)$ is also a point mass,

$$p(\boldsymbol{\epsilon}_n \,|\, \mathbf{x}_n) = \mathbb{I}[\boldsymbol{\epsilon}_n = g^{-1}(\mathbf{x}_n)],$$

where $g^{-1}$ is the left inverse of $g$. This means for injective functions of the randomness (both noise and latent variables), the "posterior" may be worth analysis as a deterministic hidden representation [1], but it is not random.

The point mass posterior can be found via nonlinear least squares. Nonlinear least squares yields the iterative algorithm

$$\hat{\boldsymbol{\epsilon}}_n = \hat{\boldsymbol{\epsilon}}_n - \rho_t \nabla_{\hat{\boldsymbol{\epsilon}}_n} f(\hat{\boldsymbol{\epsilon}}_n)^{\top} (f(\hat{\boldsymbol{\epsilon}}_n) - \mathbf{x}_n),$$

for some step size sequence $\rho_t$. Note the updates will get stuck when the gradient of $f$ is zero. However, the injective property of $f$ allows the iteration to be checked for correctness (simply check if $f(\hat{\boldsymbol{\epsilon}}_n) = \mathbf{x}_n$).

## B    Implicit Model Examples in Edward

We demonstrate implicit models via example implementations in Edward [5].

Fig. 1 implements a 2-layer deep implicit model. It uses `tf.layers` to define neural networks: `tf.layers.dense(x, 256)` applies a fully connected layer with 256 hidden units and input $x$; weight and bias parameters are abstracted from the user. The program generates $N$ data points $\mathbf{x}_n \in \mathbb{R}^{10}$ using two layers of implicit latent variables $\mathbf{z}_{n,1}, \mathbf{z}_{n,2} \in \mathbb{R}^d$ and with an implicit likelihood.

Fig. 2 implements a Bayesian GAN for classification. It manually defines a 2-layer neural network, where for each data index, it takes features $\mathbf{x}_n \in \mathbb{R}^{500}$ concatenated with noise $\boldsymbol{\epsilon}_n \in \mathbb{R}$ as input. The output is a label $\mathbf{y}_n \in \{-1, 1\}$, given by the sign of the last layer. We place a standard normal prior over all weights and biases. Running this program while feeding the placeholder $\mathbf{X} \in \mathbb{R}^{N \times 500}$ generates a vector of labels $\mathbf{y} \in \{-1, 1\}^N$.

```
1  import tensorflow as tf
2  from edward.models import Normal
3
4  # random noise is Normal(0, 1)
5  eps2 = Normal(tf.zeros([N, d]), tf.ones([N, d]))
6  eps1 = Normal(tf.zeros([N, d]), tf.ones([N, d]))
7  eps0 = Normal(tf.zeros([N, d]), tf.ones([N, d]))
8
9  # alternate latent layers z with hidden layers h
10 z2 = tf.layers.dense(eps2, 128, activation=tf.nn.relu)
11 h2 = tf.layers.dense(z2, 128, activation=tf.nn.relu)
12 z1 = tf.layers.dense(tf.concat([eps1, h2], 1), 128, activation=tf.nn.relu)
13 h1 = tf.layers.dense(z1, 128, activation=tf.nn.relu)
14 x  = tf.layers.dense(tf.concat([eps0, h1], 1), 10, activation=None)
```

**Figure 1:** Two-layer deep implicit model for data points $\mathbf{x}_n \in \mathbb{R}^{10}$. The architecture alternates with stochastic and deterministic layers. To define a stochastic layer, we simply inject noise by concatenating it into the input of a neural net layer.

```
1  import tensorflow as tf
2  from edward.models import Normal
3
4  # weights and biases have Normal(0, 1) prior
5  W1 = Normal(tf.zeros([501, 256]), tf.ones([501, 256]))
6  W2 = Normal(tf.zeros([256, 1]), tf.ones([256, 1]))
7  b1 = Normal(tf.zeros(256), tf.ones(256))
8  b2 = Normal(tf.zeros(1), tf.ones(1))
9
10 # set up inputs to neural network
11 X = tf.placeholder(tf.float32, [N, 500])
12 eps = Normal(tf.zeros([N, 1]), tf.ones([N, 1]))
13
14 # y = neural_network([x, eps])
15 input = tf.concat([X, eps], 1)
16 h1 = tf.nn.relu(tf.matmul(input, W1) + b1)
17 h2 = tf.matmul(h1, W2) + b2
18 y = tf.reshape(tf.sign(h2), [-1])  # take sign, then flatten
```

**Figure 2:** Bayesian GAN for classification, taking $\mathbf{X} \in \mathbb{R}^{N \times 500}$ as input and generating a vector of labels $\mathbf{y} \in \{-1, 1\}^N$. The neural network directly generates the data rather than parameterizing a probability distribution.

## C  KL Uniqueness

An integral probability metric measures distance between two distributions $p$ and $q$,

$$d(p, q) = \sup_{f \in \mathcal{F}} |\mathbb{E}_p f - \mathbb{E}_q f|.$$

Integral probability metrics have been used for parameter estimation in generative models [2] and for variational inference in models with tractable density [4]. In contrast to models with only local latent variables, to infer the posterior, we need an integral probability metric between it and the variational approximation. The direct approach fails because sampling from the posterior is intractable.

An indirect approach requires constructing a sufficiently broad class of functions with posterior expectation zero based on Stein's method [4]. These constructions require a likelihood function and its gradient. Working around the likelihood would require a form of nonparametric density estimation; unlike ratio estimation, we are unaware of a solution that sufficiently scales to high dimensions.

As another class of divergences, the $f$ divergence is

$$d(p, q) = \mathbb{E}_q \left[ f \left( \frac{p}{q} \right) \right].$$

Unlike integral probability metrics, $f$ divergences are naturally conducive to ratio estimation, enabling implicit $p$ and implicit $q$. However, the challenge lies in scalable computation. To subsample data in hierarchical models, we need $f$ to satisfy up to constants $f(ab) = f(a) + f(b)$, so that the expectation becomes a sum over individual data points. For continuous functions, this is a defining property of the $\log$ function. This implies the KL-divergence from $q$ to $p$ is the only $f$ divergence where the subsampling technique in our desiderata is possible.

## D  Hinge Loss

Let $r(\mathbf{x}_i, \mathbf{z}_i, \boldsymbol{\beta}; \theta)$ output a real value, as with the log loss in Section 4. The hinge loss is

$$\mathcal{D}_{\text{hinge}} = \mathbb{E}_{p(\mathbf{x}_n, \mathbf{z}_n \mid \boldsymbol{\beta})}[\max(0, 1 - r(\mathbf{x}_n, \mathbf{z}_n, \boldsymbol{\beta}; \theta))] +$$
$$\mathbb{E}_{q(\mathbf{x}_n, \mathbf{z}_n \mid \boldsymbol{\beta})}[\max(0, 1 + r(\mathbf{x}_n, \mathbf{z}_n, \boldsymbol{\beta}; \theta))].$$

We minimize this loss function by following unbiased gradients. The gradients are calculated analogously as for the log loss. The optimal $r^*$ is the log ratio.

## E  Comparing Bayesian GANs with MAP to GANs with MLE

In Section 4, we argued that MAP estimation with a Bayesian GAN enables analysis over discrete data, but GANs—even with a maximum likelihood objective [3]—cannot. This is a surprising result: assuming a flat prior for MAP, the two are ultimately optimizing the same objective. We compare the two below.

For GANs, assume the discriminator outputs a logit probability, so that it's unconstrained instead of on $[0, 1]$. GANs with MLE use the discriminative problem

$$\max_{\boldsymbol{\theta}} \mathbb{E}_{q(\mathbf{x})}[\log \sigma(D(\mathbf{x}; \boldsymbol{\theta}))] + \mathbb{E}_{p(\mathbf{x}; \mathbf{w})}[\log(1 - \sigma(D(\mathbf{x}; \boldsymbol{\theta})))].$$

They use the generative problem

$$\min_{\mathbf{w}} \mathbb{E}_{p(\mathbf{x}; \mathbf{w})}[-\exp(D(\mathbf{x}))].$$

Solving the generative problem with reparameterization gradients requires backpropagating through data generated from the model, $\mathbf{x} \sim p(\mathbf{x}; \mathbf{w})$. This is not possible for discrete $\mathbf{x}$. Further, the exponentiation also makes this objective numerically unstable and thus unusable in practice.

Contrast this with Bayesian GANs with MLE (MAP and a flat prior). This applies a point mass variational approximation $q(\mathbf{w}') = \mathbb{I}[\mathbf{w}' = \mathbf{w}]$. It maximizes the *evidence lower bound* (ELBO),

$$\max_{\mathbf{w}} \mathbb{E}_{q(\mathbf{w})}[\log p(\mathbf{w}) - \log q(\mathbf{w})] + \sum_{n=1}^{N} r(\mathbf{x}_n, \mathbf{w}).$$

The first term is zero for a flat prior $p(\mathbf{w}) \propto 1$ and point mass approximation; the problem reduces to

$$\max_{\mathbf{w}} \sum_{n=1}^{N} r(\mathbf{x}_n, \mathbf{w}).$$

Solving this is possible for discrete $\mathbf{x}$: it only requires backpropagating gradients through $r(\mathbf{x}, \mathbf{w})$ with respect to $\mathbf{w}$, all of which is differentiable. Further, the objective does not require a numerically unstable exponentiation.

Ultimately, the difference lies in the role of the ratio estimators. Recall for Bayesian GANs, we use the ratio estimation problem

$$\mathcal{D}_{\log} = \mathbb{E}_{p(\mathbf{x}; \mathbf{w})}[-\log \sigma(r(\mathbf{x}, \mathbf{w}; \boldsymbol{\theta}))] +$$
$$\mathbb{E}_{q(\mathbf{x})}[-\log(1 - \sigma(r(\mathbf{x}, \mathbf{w}; \boldsymbol{\theta})))].$$

The optimal ratio estimator is the log-ratio $r^*(\mathbf{x}, \mathbf{w}) = \log p(\mathbf{x} \mid \mathbf{w}) - \log q(\mathbf{x})$. Optimizing it with respect to $\mathbf{w}$ reduces to optimizing the log-likelihood $\log p(\mathbf{x} \mid \mathbf{w})$. The optimal discriminator for GANs with MLE has the same ratio, $D^*(\mathbf{x}) = \log p(\mathbf{x}; \mathbf{w}) - \log q(\mathbf{x})$; however, it is a constant function with respect to $\mathbf{w}$. Hence one cannot immediately substitute $D^*(\mathbf{x})$ as a proxy to optimizing the likelihood. An alternative is to use importance sampling; the result is the former objective [3].

**Figure 3: (left)** Difference of ratios over steps of $q$. Low variance on $y$-axis means more stable. Interestingly, the ratio estimator is more accurate and stable as $q$ converges to the posterior. **(middle)** Difference of ratios over steps of $r$; $q$ is fixed at random initialization. The ratio estimator doesn't improve even after many steps. **(right)** Difference of ratios over steps of $r$; $q$ is fixed at the posterior. The ratio estimator only requires few steps from random initialization to be highly accurate.

# F   Stability of Ratio Estimator

With implicit models, the difference from standard KL variational inference lies in the ratio estimation problem. Thus we would like to assess the accuracy of the ratio estimator. We can check this by comparing to the true ratio under a model with tractable likelihood.

We apply Bayesian linear regression. It features a tractable posterior which we leverage in our analysis. We use 50 simulated data points $\{\mathbf{y}_n \in \mathbb{R}^2, \mathbf{x}_n \in \mathbb{R}\}$. The optimal (log) ratio is

$$r^*(\mathbf{x}, \boldsymbol{\beta}) = \log p(\mathbf{x} \mid \boldsymbol{\beta}) - \log q(\mathbf{x}).$$

Note the log-likelihood $\log p(\mathbf{x} \mid \boldsymbol{\beta})$ minus $r^*(\mathbf{x}, \boldsymbol{\beta})$ is equal to the empirical distribution $\sum_n \log q(\mathbf{x}_n)$, a constant. Therefore if a ratio estimator $r$ is accurate, its difference with $\log p(\mathbf{x} \mid \boldsymbol{\beta})$ should be a constant with low variance across values of $\boldsymbol{\beta}$.

See Fig. 3. The top graph displays the estimate of $\log q(\mathbf{x})$ over updates of the variational approximation $q(\boldsymbol{\beta})$; each estimate uses a sample from the current $q(\boldsymbol{\beta})$. The ratio estimator $r$ is more accurate as $q$ exactly converges to the posterior. This matches our intuition: if data generated from the model is close to the true data, then the ratio is more stable to estimate.

An alternative hypothesis for Fig. 3 is that the ratio estimator has simply accumulated information during training. This turns out to be untrue; see the bottom graphs. On the left, $q$ is fixed at a random initialization; the estimate of $\log q(\mathbf{x})$ is displayed over updates of $r$. After many updates, $r$ still produces unstable estimates. In contrast, the right shows the same procedure with $q$ fixed at the posterior. $r$ is accurate after few updates.

Several practical insights appear for training. First, it is not helpful to update $r$ multiple times before updating $q$ (at least in initial iterations). Additionally, if the specified model poorly matches the data, training will be difficult across all iterations.

The property that ratio estimation is more accurate as the variational approximation improves is because $q(\mathbf{x}_n)$ is set to be the empirical distribution. (Note we could subtract any density $q(\mathbf{x}_n)$ from the ELBO in Equation 4.) Likelihood-free variational inference finds $q(\boldsymbol{\beta})$ that makes the observed data likely under $p(\mathbf{x}_n \mid \boldsymbol{\beta})$, i.e., $p(\mathbf{x}_n \mid \boldsymbol{\beta})$ gets closer to the empirical distribution at values sampled from $q(\boldsymbol{\beta})$. Letting $q(\mathbf{x}_n)$ be the empirical distribution means the ratio estimation problem will be less trivially solvable (thus more accurate) as $q(\boldsymbol{\beta})$ improves.

Note also this motivates why we do not subsume inference of $p(\boldsymbol{\beta} \mid \mathbf{x})$ in the ratio in order to enable implicit global variables and implicit global variational approximations. Namely, estimation requires comparing samples between the prior and the posterior; they rarely overlap for global variables.