[Reviews · NeurIPS 2017]

Reviewer 1



The paper presents an implicit variational inference method for likelihood-free inference. This approach builds on previous work and particularly on Hierarchical Variational Inference and Implicit Variational Bayes. The key trick used in the paper is the subtraction of the log empirical distribution log q(xn) and the transformation of the ELBO in the form given by eq. 4, which suggests the use of log density ratio estimation as a tool for likelihood-free variational inference. The rest methodological details of the papers are based on standard tools, such as log density ratio estimation, reparametrization and hierarchical variational distributions. While I found the trick to deal with likelihood intractability very interesting, it requires log density ratio estimation in high-dimensional spaces (in the joint space of data x_n and latent variable z_n). This is very challenging since log density ratio estimation in high dimensions is an extremely difficult problem and there is no clear evidence that the authors provide a stable algorithm to deal that. For instance, the fact that the authors have not applied their method to a standard GAN (for generating high dimensional data such as images) but instead they have constructed this rather weird Bayesian GAN for classification (see page 7) indicates that the current algorithm is very unstable. In fact it is hard to see how to stabilize the proposed algorithm since initially the “variational joint” will be very different from the "real joint" and it is precisely this situation that makes log density ratio estimation completely unreliable, leading to very biased gradients in the early crucial iterations of the optimization.

Reviewer 2



The paper defines a class of probability models -- hierarchical implicit models -- consisting of observations with associated 'local' latent variables that are conditionally independent given a set of 'global' latent variables, and in which the observation likelihood is not assumed to be tractable. It describes an approach for KL-based variational inference in such 'likelihood-free' models, using a GAN-style discriminator to estimate the log ratio between a 'variational joint' q(x, z), constructed using the empirical distribution on observations, and the true model joint density. This approach has the side benefit of supporting implicit variational models ('variational programs'). Proof-of-concept applications are demonstrated to ecological simulation, a Bayesian GAN, and sequence modeling with a stochastic RNN. The exposition is very clear, well cited, and the technical machinery is carefully explained. Although the the application of density ratio estimation to variational inference seems to be an idea 'in the air' and building blocks of this paper have appeared elsewhere (for example the Adversarial VB paper), I found this synthesis to be cleaner, easier to follow, and more general (supporting implicit models) than any of the similar papers I've read so far. The definition of hierarchical implicit models is a useful point in theoretical space, and serves to introduce the setup for inference in section 3. However the factorization (1), which assumes iid observations, is quite restrictive -- I don't believe it technically even includes the Lotka-Volterra or stochastic RNN models explored in the paper itself! (since both have temporal dependence). It seems worth acknowledging that the inference approach in this paper is more general, and perhaps discussing how it could be adapted to problems and models with more structured (time series, text, graph) observations and/or latents. Experiments are probably the weakest point of this paper. The 'Bayesian GAN' is a toy and the classification setup is artificial; supervised learning is not why people care about GANs. The symbol generation RNN is not evaluated against any other methods and it's not clear it works particularly well. The Lotka-Volterra simulation is the most compelling; although the model has few parameters and no latent variables, it nicely motivates the notion of implicit models and shows clear improvement on the (ABC) state of the art. Overall there are no groundbreaking results, and much of this machinery could be quite tricky to get working in practice (as with vanilla GANS). I wish the experiments were more compelling. But the approach seems general and powerful, with the potential to open up entire new classes of models to effective Bayesian inference, and the formulation in this paper will likely be useful to many reasearchers as they begin to flesh it out. For that reason I think this paper is a valuable contribution. Misc comments and questions: Lotka-Volterra model: I'm not sure the given eqns (ln 103) are correct. Shouldn't the Beta_3 be added, not subtracted, to model the predator birth rate? As written, dx_2/dt is always negative in expectation which seems wrong. Also Beta_2 is serving double duty as the predator *and* prey death rate, is this intentional? Most sources (including the cited Papamakarios and Murray paper) seem to use four independent coefficients. line 118: "We described two classes of implicit models" but I only see one? (HIMs) line 146: "log empirical log q(x_n)" is redundant Suppose we have an implicit model, but want to use an explicit variational approximation (for example the mean-field Gaussian in the Lotka-Volterra experiment). Is there any natural way to exploit the explicit variational density for faster inference? Subtracting the constant log q(x) from the ELBO means the ratio objective (4) no longer yields a lower bound to the true model evidence; this should probably be noted somewhere. Is there an principled interpretation of the quantity (4)? It is a lower bound on log p(x)/q(x), which (waving hands) looks like an estimate of the negative KL divergence between the model and empirical distribution -- maybe this is useful for model criticism?

Reviewer 3



Thank you for an interesting read. This paper proposed a hierarchical probabilistic model using implicit distributions. To perform posterior inference the authors also proposed a variational method based on GAN-related density ratio estimation techniques. The proposed method is evaluated with a number of different tasks including ABC, supervised learning and generative modeling. I like the idea in general but I think there are a few points that need to be made clearer. 1. How is your method related to AVB [36] and ALI [13]? I can see these connections, but not all the readers you're targeting could see it easily. 2. In the AVB paper they mentioned a crucial trick (adaptive contrast) to improve the density ratio estimations in high dimensions. You only did a toy case (2D linear regression) to demonstrate the stability of your method, and your findings are essentially the same as in the toy example in the AVB paper (naive density ratio estimation works well in low dimensional case). It would be better if you could provide an analysis in high dimensional case, e.g. your BNN example. 3. Hinge loss: why the optimal r is the log ratio? 4. Generating discrete data: yes using r(x, w) instead of r(x) could provide gradients, however this means you need to input w to the discriminator network as well. Usually you need quite a deep (and wide) neural network to generate realistic data so I presume w could be of very high dimensions. How scalable is your approach here? 5. I'm a bit worried about no quantitative results for the sequence generation part. I think it's not a good practice for just including generative samples and letting the readers judge the fidelity. In summary I think this paper is borderline. I would be happy to see clarifications if the authors think I've missed some important points.